# The Role of Cytokines in Orthodontic Tooth Movement

**DOI:** 10.3390/ijms26146688

**Published:** 2025-07-11

**Authors:** Hideki Kitaura, Fumitoshi Ohori, Aseel Marahleh, Jinghan Ma, Angyi Lin, Ziqiu Fan, Kohei Narita, Kou Murakami, Hiroyasu Kanetaka

**Affiliations:** 1Division of Orthodontics and Dentofacial Orthopedics, Tohoku University Graduate School of Dentistry, 4-1, Seiryo-machi, Aoba-ku, Sendai 980-8575, Miyagi, Japan; fumitoshi.ohori.t3@dc.tohoku.ac.jp (F.O.); ma.jinghan.c1@tohoku.ac.jp (J.M.); lin.angyi.r5@dc.tohoku.ac.jp (A.L.); fan.ziqiu.q1@dc.tohoku.ac.jp (Z.F.); kohei.narita.a2@tohoku.ac.jp (K.N.); kou.murakami.b2@tohoku.ac.jp (K.M.); hiroyasu.kanetaka.e6@tohoku.ac.jp (H.K.); 2Frontier Research Institute for Interdisciplinary Sciences, Tohoku University, 6-3, Aramaki Aza, Aoba Aoba-ku, Sendai 980-8575, Miyagi, Japan; aseel.mahmoud.suleiman.marahleh.e6@tohoku.ac.jp; 3Division of Advanced Dental Science and Technology, Graduate School of Biomedical Engineering, Tohoku University, 6-6-12, Aramaki Aza, Aoba Aoba-ku, Sendai 980-8579, Miyagi, Japan

**Keywords:** orthodontic tooth movement, cytokine, osteoclast, bone

## Abstract

A challenge in orthodontic treatment is the long time taken to move teeth, which extends the long treatment period. Accordingly, various treatment protocols and orthodontic materials have been developed to shorten the orthodontic treatment period. However, controlling biological reactions is considered necessary to further shorten this treatment period. Orthodontic force results in compression of the periodontal ligament in the direction of tooth movement, resulting in various reactions in the periodontal ligament that induce osteoclast development, alveolar bone absorption, and teeth movement. The aforementioned reactions include immune reactions. Cytokines are substances responsible for intercellular communication and are involved in various physiological actions, including immune and inflammatory reactions. They cause various cellular responses, including cell proliferation, differentiation, cell death, and functional expression. Various cytokines are involved in biological reactions during orthodontic tooth movement (OTM). It is important to understand the role of cytokines during OTM in order to elucidate their biological response. This review discusses the role of cytokines during OTM.

## 1. Introduction

Orthodontic tooth movement (OTM), which is achieved by applying orthodontic mechanical force to teeth, involves reorganization of the alveolar bone and periodontal ligament (PDL). During this process, immune reactions occur in the PDL and alveolar bone, playing an essential role in initiating and regulating bone remodeling.

OTM is crucially involved in the treatment of malocclusion [1,2]. Mechanical loading induces tooth displacement, which creates a PDL compression area in the direction of force action and a tension area on the other side [3]. This causes localized aseptic inflammation, which induces bone resorption and formation at the compression and tension areas, respectively (Figure 1) [4]. The PDL is the primary structure that senses and transmits orthodontic forces from the teeth to the alveolar bone [5]. In addition to bone resorption and formation, the extracellular matrix of the PDL is continuously remodeled during OTM, which is primarily regulated by the cellular components of the PDL [6,7]. Mechanical stimulation induces biological responses in numerous cell types within the PDL and alveolar bone. It is important to elucidate the highly complex cellular mechanisms underlying OTM, especially the involved immune reactions [8].

Osteoclasts are cells found on the bone surface during resorption and are therefore crucially involved in OTM. Macrophage colony-stimulating factor (M-CSF) and receptor activator of NF-κB ligand (RANKL) promote the differentiation of osteoclast precursor cells into bone-resorbing osteoclasts [9]. Additionally, tumor necrosis factor-α (TNF-α) induces osteoclast formation [10].

Cytokines monitor and control inflammatory and immune reactions through complicated networks; moreover, they serve as biomarkers for numerous disease types [11]. Additionally, cytokines control the maturation, growth, responsiveness, cytotoxic immunity, allergic immunity, and eosinophilia of specific cell populations [12]. Cytokines are central mediators of the biological cascade during OTM. When orthodontic force compresses the PDL on one side and stretches it on the other, various cytokines are secreted in response to mechanical stress [3]. Many cytokines are involved in the recruitment and activation of immune cells at the site of force application, thereby initiating and regulating bone remodeling during OTM [13,14]. They regulate osteoclast differentiation and bone resorption, with some promoting osteoclastogenesis and others acting as inhibitors [15]. Following the active phase of bone resorption, cytokines also participate in tissue repair and regeneration during OTM. Some cytokines, such as insulin-like growth factor (IGF) and bone morphogenetic proteins (BMPs), are involved in promoting osteoblast differentiation, angiogenesis, and bone formation, thereby facilitating the remodeling of periodontal tissues [16,17]. Importantly, the timing, balance, and local concentration of cytokine release are critical to ensure efficient and controlled OTM while avoiding adverse tissue destruction. Cytokine expression is dynamic and varies during different stages of OTM, reflecting the tightly regulated interplay between inflammatory activation and resolution phases.

Cytokine production in the gingival area, including gingival crevicular fluid (GCF), during OTM has been extensively studied to elucidate the cell metabolism at the local site as well as the periodontal tissue health and bone remodeling. Accordingly, cytokines may play a regulatory role in osteoclast formation. This review describes and discusses the role of OTM-related cytokines, with a focus on their effects on osteoclasts.

## 2. Cytokines Involved in Orthodontic Tooth Movement

In this review, we will describe the characteristics of the following cytokines; Interleukin (IL)-1, IL-2, IL-4, IL-5, IL-6, IL-7, IL-8, IL-10, IL-11, IL-12, IL-13, IL-16, IL-17, IL-18, IL-20, IL-23, IL-27, IL-33, IL-34, IFN-γ, TNF-α, M-CSF, TGFs, RANKL, and OPG, as well as their relationship to the immune system, bone metabolic diseases and their effects on osteoclast formation and OTM.

### 2.1. IL-1β

IL-1β is an important inflammatory cytokine produced by macrophages, T cells, monocytes, and dendritic cells [18]. It promotes osteoclast formation and bone resorption in rheumatoid arthritis, osteoporosis, and periodontal disease [6,19,20,21]. However, it cannot solely induce osteoclast differentiation from osteoclast precursor cells. Nonetheless, IL-1RI induction through c-Fos or IL-1RI receptor overexpression in bone marrow stromal cells can induce true osteoclast formation through a RANKL/RANK-independent mechanism [22]. IL-1 is not considered a mere substitute for RANKL, with reports suggesting that the in vivo effect of IL-1 on osteoclasts depends on its effect on osteoblasts [23].

As aforementioned, the PDL is crucially involved in the bone remodeling process by promoting bone resorption and formation in the compression and tension areas, respectively, in response to mechanical force during OTM. Notably, these processes are regulated by IL-1 [24]. The application of orthodontic forces to teeth has been shown to elevate IL-1β levels in GCF [25,26]. IL-1β is an important mediator in various acute-phase inflammatory and immune responses [27]; further, it is strongly involved in the periodontal tissue reorganizing process during the early stage of OTM [28,29]. Specifically, IL-1β prompts osteoclast differentiation, enhances their fusion and survival, and activates their function. Application of traction forces to the maxillary canine increases IL-1β levels in GCF on the distal tooth surface, peaking at 24–72 h after tooth movement initiation [30]. Additionally, the OTM rate is positively correlated with IL-1β levels in GCF [31]. The effect of IL-1β is regulated by its receptor antagonist, interleukin-1 receptor antagonist (IL-1RA); further, orthodontic mechanical force has been associated with an increase in both IL-1β secretion and the IL-1β/IL-1RA ratio [32]. IL-1, which is among the most potent cytokines in the early stage of OTM, induces cell proliferation and differentiation; further, it is secreted by various cell types such as fibroblasts, macrophages, cementoblasts, osteoblasts, and osteoclasts in response to external stimulation [28,33]. IL-1β may serve as a biomarker for the extent of OTM, which is dependent on the efficiency of alveolar bone remodeling [34]. At 24 h after orthodontic treatment initiation, there is an increase in both pocket fluid volume and IL-1 levels. Here, IL-1 levels in pocket fluid differed between adolescents and young adults, suggesting age-dependent differences in the tissue response. As a result, teeth have quicker mobility in teenagers than in young adults [35]. A study on the effects of low-level laser therapy on cytokine levels during OTM found that photobiomodulation was significantly related to elevated IL-1b levels in GCF [36]. Further, adolescents with obesity have shown higher orofacial pain severity at 24 h after OTM, as well as higher IL-1β levels before and during OTM, compared with adolescents without obesity [26]. Compared with the control group, rats who underwent ligature appliance as experimental OTM showed a greater increase in gene expression of IL-1β. Further, the ligation device group had greater linear bone resorption, bone mineral density, and trabecular number; lower values of fractional bone mass; and increased values of trabecular spacing compared with the other groups [37]. Autophagy inhibition with 3-methyladenine was shown to accelerate OTM and reduce bone mineral density, as well as increase the entropy of the PDL structure; contrastingly, autophagy enhancement with rapamycin yielded opposite results in OTM. In addition, there was a post-OTM increase in IL-1 levels, which was highest in the group treated with 3-methyladenine (autophagy inhibition) [38]. Taken together, IL-1 appears to directly enhance osteoclast formation and indirectly induce osteoclastogenic cytokines in OTM.

### 2.2. IL-2

IL-2 is a T helper 1 (Th1) cell-derived pro-inflammatory cytokine that is involved in macrophage stimulation, B cell activation, natural killer (NK) cell and T cell proliferation, and stimulation of osteoclast activity. Further, IL-2 is involved in the stimulation of osteoclast activity in bone resorption, which suggests an active pathological involvement in periodontal disease [39]. Contrastingly, low-dose IL-2 has been shown to attenuate osteoclast formation in collagen-induced arthritis via a c-Jun N-terminal kinase (JNK)-dependent pathway [40]. Notably, IL-2 polymorphism has been associated with the severity of chronic periodontitis [41]. A human study showed that during OTM, IL-2 levels were increased in GCF [42].

### 2.3. IL-4

IL-4 is primarily expressed in T helper 2 (Th2) cells, eosinophils, mast cells, basophils, and other immune cells; further, it plays a crucial role in regulating immune responses [43]. In addition, it plays a crucial role in allergic inflammation and parasitic infections, as well as stimulates B cell proliferation and the activation of eosinophils, basophils, and mast cells. IL-4 can antagonize Th1-induced inflammatory immune responses suppress the synthesis of numerous inflammatory cytokines [44,45]. Further, IL-4 prevents lipoprotein-induced periodontitis by inhibiting osteoclast formation [46]. IL-4 is a strong inhibitor of the RANKL-induced osteoclastogenic process [47]. Furthermore, IL-4 has been shown to inhibit TNF-α-induced osteoclast formation in vitro [48], as well as through RANKL expression by TNF-α-activated stromal cells and TNF-α-activated osteoclast precursors in vivo [49].

A mouse model of tooth movement showed that IL-4 inhibited OTM and osteoclast formation [50]. In a rat model of OTM, injection of exogenous IL-4 into the local site following tooth movement reduced relapse by inhibiting osteoclast formation [51]. In vitro cultivated primary osteoblasts from mouse calvariae that were subjected to micro-pulse vibration for 20 min showed an increase in IL-4 levels [52]. Accordingly, analyzing plasma proteins may facilitate the prediction and monitoring of the dynamics of bone and tissue remodeling, including during orthodontic treatment. A study applied two different orthodontic forces (low force and normal force) for bilateral buccal expansion around the maxillary second and third molars of rats. Notably, they observed no significant intergroup differences in the IL-4 levels [53].

### 2.4. IL-5

IL-5 is a type 2 cytokine produced by several immune cells, including Th2 cells, eosinophils, mast cells, and basophils [54]. Under normal conditions, IL-5 transgenic mice showed an increase in the number of eosinophils and the bone mass; further, IL-5 protects against inflammation- and hormone-induced bone loss [55]. Accordingly, IL-5 suppresses osteoclast formation.

GCF samples were taken 10 weeks after initial appliance placement at 4 h, 7 days, and 42 days after distal force was applied to the maxillary canines. IL-5 levels were measured in GCF using multiplex assays. However, the level of IL-5 was undetectable in the samples [56].

### 2.5. IL-6

IL-6 is a pro-inflammatory and pleiotropic cytokine that regulates the immune system [57]. It is crucially involved in inflammation, diabetes, atherosclerosis, hematopoiesis, autoimmunity, cancer, trauma, and rheumatoid arthritis [58,59,60,61,62]. It is released by several cell types, including B cells, T cells, monocytes, macrophages, endothelial cells, adipocytes, mesangial cells, fibroblasts, keratinocytes, and several tumor cells [63]. IL-6 indirectly affects osteoclasts; further, it promotes osteoclast formation and bone resorption by enhancing RANKL expression in IL-6-activated osteoblasts [64,65]. IL-6-neutralizing antibodies attenuate TNF-α- and IL-1β-induced osteoclast formation [66]. Accordingly, IL-6 promotes osteoclast bone resorption and plays a crucial role in the etiology of bone loss in acute and chronic inflammation [67], periodontitis [21], osteoporosis, and rheumatoid arthritis [64].

In a previous study, the maxillary canine that underwent distal movement was used as the experimental tooth, while the opposite canine was used as the control. Here, the IL-6 levels in GCF were higher in the experimental tooth than in the control tooth [68]. Adolescents have shown higher IL-6 levels in crevicular fluid than young adults [35]. Additionally, periodontitis induced elevated gingival IL-6 and CXCL2 levels in a rat model of tooth movement. Further, OTM promoted bacterial-induced periodontal tissue destruction and IL-6 gene expression in the gingiva. Similarly, increased gingival IL-6 and CXCL2 levels were observed in human periodontitis [69]. Compared with normal-weight individuals, individuals with obesity showed higher IL-6 levels in GCF during distal movement of canines [70]. Additionally, increased IL-6 levels have been reported in the GCF of individuals undergoing orthodontic treatment with clear aligners [71]. IL-6 signaling was activated in the PDL following orthodontic intervention in a rat orthodontic model. This signaling promoted OTM by inducing osteoclast bone resorption; further, IL-6 increased the number of osteoclasts by suppressing apoptosis as well as enhancing responsiveness to M-CSF and RANKL. Moreover, IL-6 signaling induces neuroinflammation in the trigeminal ganglion, which results in orthodontic pain. Taken together, IL-6 signaling regulates both tooth movement and pain during OTM [72]. Local static magnetic field stimulation has significantly increased the tooth movement distance and induced osteoclast formation in the compressed side of a rat model of OTM. IL-6 levels were increased in force-loaded PDL stem cells exposed to static magnetic field stimulation, as well as in the compressed side of a rat model of OTM. The OTM distance enhanced by static magnetic field stimulation was significantly reduced by administration of tocilizumab, which is an IL-6 inhibitor. Taken together, static magnetic field stimulation appears to induce IL-6 secretion by force-loaded PDL stem cells, which promotes OTM and osteoclast formation [73]. Exogenous application of direct current to piezoelectric biopolymers induces biochemical changes in the intra- and extracellular regions, significantly affecting the bone metabolism rate. Further, micro-current stimulation increased the rate of tooth movement and IL-6 levels in rats [74]. There were dynamic fluctuations in IL-6 and miR-146a expression patterns during orthodontic relapse in a rat model. Specifically, IL-6 protein levels peaked on day 7 following removal of the orthodontic appliance, which was accompanied by a decrease in miR-146a expression. In vitro studies have shown that miR-146a inhibition resulted in elevated IL-6 expression, indicating its regulatory function [75].

### 2.6. IL-7

IL-7 is expressed by stromal cells; further, it is crucially involved in the homeostatic survival and development of naive T cells, memory T cells, immature thymocytes, pro-B cells, and innate lymphoid cells. It contains four antiparallel α-helices that bind to type I cytokine receptors. Given this property, pharmacological treatment with IL-7 induces proliferation of pro-B cells, naive T cells, and memory T cells [76]. IL-7 induces bone loss through the induction of RANKL and TNF-a from T cells in vivo [77,78]. In rheumatoid arthritis, M1 macrophages show increased IL-7R expression, which increases their responsiveness to IL-7-induced osteoclast formation [79]. Notably, IL-7 enhances the generation of osteoclast precursors in vitro. Ovariectomized mice showed increased IL-7, but not RANKL, mRNA expression by osteoblasts and stromal cells [80]. Notably, patients wearing aligners did not show significant changes in IL-7 levels in GCF samples obtained from the lower incisors [81].

### 2.7. IL-8

IL-8, which is also known as CXCL8, is a CXC chemokine crucially involved in regulating inflammatory responses [82]. Further, it is a neutrophil chemotactic factor. IL-8 is synthesized by various cells, including macrophages, endothelial cells, airway smooth muscle cells, and epithelial cells [83]. IL-8 induces its effects by binding to its receptor (CXCR1 or CXCR2) [84]. IL-8 is expressed in numerous tumors and cancer cell lines; further, it promotes angiogenesis, tumor metastasis, and tumor growth in various human cancers [85]. Additionally, IL-8 levels are increased in the synovial fluid and serum of patients with rheumatoid arthritis [86]. IL-8 triggers RANKL expression in bone marrow stromal cells [87].

The study compared IL-8 levels in GCF during OTM using self-ligating brackets and conventional brackets. A study reported higher IL-8 levels in GCF in patients with conventional brackets than in those with self-ligating brackets [88]. In a rat OTM model, orthodontic compressive force enhanced IL-8 mRNA and protein expression from PDL cells in a force-dependent manner [89]. Another study on the effects of mechanical stress on inflammatory remodeling of periodontal tissues and IL-8 expression found that both inflammatory stimuli and orthodontic force induced IL-8 expression [90]. Orthodontic treatment with clear aligners significantly increased IL-8 levels in GCF [71]. Photobiomodulation increased tooth movement by modulating IL-8 levels, which were higher compared with those in the non-irradiated area. [36,91]. Orthodontic forces induced changes in IL-8 levels in human periodontal tissue. Following mechanical stimulation, there were differences in IL-8 levels at both tension and pressure sites, which may trigger the bone remodeling process. Early orthodontic forces significantly increased IL-8 levels after 1 h, which peaked on Day 6. Finally, local host responses to orthodontic forces have been shown to increase the accumulation of IL-8 and neutrophils, which may trigger the bone remodeling process [92].

### 2.8. IL-10

IL-10 is an anti-inflammatory cytokine crucially involved in suppressing Th1 cell responses by suppressing the expression and function of various inflammatory cytokines, including interferon gamma (IFN-γ), TNF, IL-1, and IL-6, in monocytes and T cells. IL-10 promotes immune events, including the cytotoxic activity of CD8+ T cells and NK cells, thymocyte proliferation, and immunoglobulin production by B cells [93,94,95]. Therefore, IL-10 plays both stimulatory and inhibitory roles in immune responses; further, IL-10 inhibits osteoclast differentiation in vitro and in vivo [96,97,98,99]. Additionally, IL-10 inhibits osteolysis in periodontitis and bone loss diseases [97,100,101].

A study reported a non-significant increase in IL-10 levels in GCF on day 7 after orthodontic treatment activation, which subsequently dissipated [102]. Moreover, IL-10 cytokine levels in GCF did not differ between patients treated with fixed orthodontics and those treated with Invisalign [103]. Compared with controls, mice treated with IL-1Ra showed decreased OTM and TRAP-positive osteoclasts, as well as increased IL-10 levels in the periodontal tissues [104]. An in vitro experiment showed that compressive force causes M2 polarization of macrophages, which increased IL-10 gene expression, via H3 histone acetylation [105].

### 2.9. IL-11

IL-11 is a multifunctional cytokine expressed in various tissues and cells, including epithelial cells, osteoclasts, osteoblasts, fibroblasts, hematopoietic cells, mesenchymal stem cells, central nervous system neurons, synovial cells, adipocytes, gastrointestinal tract, and chondrocytes. IL-11 belongs to the IL-6 cytokine family and mainly exists as a quadruplex structure. IL-11 induces various immune activities in both innate and adaptive immunity. IL-11 can directly regulate macrophage activity by inhibiting IL-1β, IL-12, and TNF-α, which are pro-inflammatory cytokines, in vitro. Taken together, IL-11 acts as an anti-inflammatory cytokine by regulating macrophage functions [106,107]. IL-11 is produced by bone marrow stromal cells. Further, IL-11 induces osteoclast formation and indirectly acts on osteoclast precursors. Moreover, IL-11 has been shown to induce bone destruction in a mouse calvarial bone organ culture model [108,109]. IL-11 is involved in several osteolytic bone diseases, such as rheumatoid arthritis and osteoporosis. Both diseases involve significantly increased serum IL-11 levels, which are positively correlated with the levels of bone resorption markers [110,111].

Patients undergoing orthodontic treatment with compressive force showed significantly increased IL-11 levels in the compressed PDL [112]. Another study found that vibration stimulation during OTM increased movement and IL-11 levels compared with the control group [113].

### 2.10. IL-12

IL-12 is a pro-inflammatory cytokine that is primarily produced by dendritic cells and phagocytes. IL-12 induces the conversion of naive CD4+ T cells to Th1 cells and induces IFN-γ production in Th1 cells [114]. IL-12 increases MHC I and MHC II expression on tumor cells, which promotes their recognition and lysis [115]. IL-12 exerts remarkable antitumor effects dependent on NK cells, NK T cells, and CD8+ T cells [116]. It has two subunits, p35 and p40, which form the p70 heterodimer via three disulfide bonds [117]. IL-12 was found to reduce RANK-induced osteoclast formation and RANKL interaction via a T cell-independent process [118]. Moreover, TNF-α-induced osteoclast formation was inhibited by apoptosis induction through the interaction of IL-12-induced FasL in non-adherent cells with TNF-α-induced Fas in adherent cells in bone marrow cell cultures in vitro [119]. IL-12 can inhibit osteoclast formation in spleen cell cultures in vitro via a T cell-dependent process [120].

In a mouse study on the effect of IL-12 on OTM, IL-12 was locally administered near the first molar every second day. IL-12-treated mice showed reduced OTM and increased apoptotic cells in the pressure area, which suggests that IL-12 attenuates OTM. This phenomenon can be attributed to IL-12-induced apoptosis of osteoclasts [121].

### 2.11. IL-13

IL-13 is expressed as a preform by granulocytes, including mast cells, basophils, and eosinophils; further, it is involved in immunoglobulin regulation, inflammation, antiparasitic responses, fibrosis, and allergic responses, with IgE being crucially involved [122]. IL-13 acts as a messenger in immune processes, especially in the induction of allergic responses [123]. IL-13 impairs maintenance of barrier function and wound healing [124]. Further, IL-13 exerts protective effects against bone destruction by suppressing bone resorption in mouse calvaria in vivo and inhibiting osteoclast formation by bone marrow macrophages and spleen cells in vitro [125]. IL-13 injection increases osteoprotegerin (OPG) expression and reduces bone destruction in the joints of mice with collagen-induced arthritis [126]. Primary osteoblasts obtained from mouse calvariae that were cultivated in vitro and subjected to micro-pulse vibration showed increased IL-13 levels [52].

In patients with fixed orthodontic appliances, there was a slight increase in IL-13 levels in GCF at 7 days after treatment initiation, which subsequently dissipated [102].

### 2.12. IL-16

IL-16 is a pro-inflammatory cytokine primarily known for its chemotactic properties, meaning it attracts specific immune cells, including CD4+ lymphocytes, monocytes, and eosinophils, to sites of inflammation or infection [127]. IL-16 directly induces differentiation of monocytes into osteoclasts through JNK/MAPK signaling-dependent activation of NFATc1 [128]. A previous study showed no significant difference between IL-16 levels in GCF between patients who underwent invisible and fixed orthodontic treatment [129].

### 2.13. IL-17

The IL-17 family, which comprises IL-17A (IL-17), IL-17B, IL-17C, IL-17D, IL-17E, and IL-17F, are crucially involved in regulating inflammation and the immune system [130]. IL-17A (commonly known as IL-17) is the most widely studied cytokine due to its pro-inflammatory actions. IL-17 is primarily produced by T helper 17 (Th17) cells, as well as other immune cells such as gamma-delta T cells, NK cells, NK T cells, macrophages, B cells, innate lymphocytes, mast cells, and neutrophils [131]. IL-17 activates multiple downstream signaling cascades, including ERK1/2, JNK, and p38, as well as NF-κB, STAT3, and Nrf2/keap1 [130]. IL-17 induces the expression of several pro-inflammatory cytokines and chemokines, including IL-1β, IL-6, IL-8, IL-23, TNF-α, CCL4, CCL20, CXCL1, and CXCL12 [132]. Th17 cells exert a direct or indirect osteoclastogenic effect via IL-17-induced activation of osteoclast-related molecules in various target cells. IL-17 treatment of CD14+ cells enhances the expression of osteoclast-related genes, including c-fms, TRAP, and RANK, which leads to an increase in the number of osteoclasts [133]. Furthermore, IL-17 directly promotes osteoclast formation and CD11b+ cell activation by increasing TNF-α and RANKL expression from IL-17-treated monocytes, even in the absence of osteoblasts or exogenous sRANKL [134].

In Wistar rats subjected to orthodontic force, there was a significant increase in IL-17-positive odontoclasts in the jiggling force during OTM [135]. Further, there was IL-17-induced activation of osteoclastogenesis and odontoclastogenesis in the rat OTM model. Taken together, the activation of Th17 cells may exacerbate root resorption through cytokine production upon exertion of excessive orthodontic forces to the PDL [136]. During orthodontic treatment, the levels of 1-25-dihydroxycholecalciferol and salivary cytokine IL-17A showed a negative correlation at various stages of OTM. This suggests that low vitamin D levels may extend the treatment duration and that vitamin D supplementation may be clinically useful in such patients [137]. IL-17 levels in human GCF were significantly up-regulated during tooth movement. Furthermore, IL-17 was positively correlated with MMP-1, MMP-2, MMP-3, MMP-8, MMP-9, and MMP-13 expression [138]. During OTM, there were significantly increased IL-17A and IL-17F levels in the GCF, which were positively correlated with RANKL expression. This suggests that Th17 cytokines are crucially involved in regulating OTM [139].

### 2.14. IL-18

IL-18, which is a stimulator of IFN-γ production by T cells and NK cells, is a pro-inflammatory cytokine crucially involved in regulating immune responses [140]. IL-18 is primarily expressed by activated macrophages and dendritic cells [141]. IL-18 plays an important role in several biological processes, including cell growth, innate and adaptive immunity, and inflammation regulation [142]. IL-18 induces its biological effects by binding to the IL-18 receptor, which is expressed on various cells, including B cells, T cells, NK cells, and macrophages [143]. Upon binding, it stimulates downstream signaling pathways, including the NF-κB and MAPKs pathways, which induce the synthesis of various cytokines and chemokines [144]. There is decreased RANKL-induced osteoclast formation in the presence of IL-18 in vitro [120]. Moreover, IL-18 inhibited TNF-α-induced osteoclast formation in bone marrow cell cultures through the interaction between TNF-α-induced Fas in adherent cells and IL-18-induced FasL in non-adherent cells [145]. IL-12 and IL-18 have a synergistic inhibitory effect on TNF-α-mediated osteoclast formation [146], which involves upregulation of FasL on non-adherent cells in bone marrow cell cultures [145]. In vivo, this effect was shown to involve a T cell-independent mechanism [147].

There was an increased IL-18 gene expression in the rat experimental tooth movement model [34]. Vibration stimulation has been shown to increase the levels of osteoclast biomarkers such as RANKL and RANKL/OPG, as well as inflammatory markers such as IL-18; moreover, it significantly improved tooth mobility and GCF volume [113]. Rats with ligature-induced periodontal disease with OTM have shown significantly higher IL-18 gene and protein expression than those without OTM. Orthodontic force loading regulates the inflammatory reactions in periodontal disease by up-regulating the production of various pro-inflammatory mediators, including IL-18 and its receptors, which increases bone resorption [148].

### 2.15. IL-20

IL-20 is a pleiotropic pro-inflammatory cytokine and an IL-10 family member that is produced in endothelial cells, epithelial cells, and monocytes [149]. IL-20 differentially regulates preosteoclast proliferation and apoptosis. Bone mesenchymal stem cells cultured in conditioned medium with IL-20 showed significantly enhanced osteoclast formation and bone resorption. This further demonstrates that IL-20 differentially regulates bone mesenchymal stem cells in osteoclastogenesis and exerts its effect by binding RANKL and RANK, as well as through the NF-κB, MAPK, and AKT signaling pathways [150]. Additionally, IL-20 is associated with periodontitis, with IL-20 contributing to periodontitis through osteoclast formation and collagen degradation [151].

In a rat model of OTM, intraperitoneal injection of IL-20 significantly increased the OTM rate and markedly activated the mechanical stress-sensing protein YAP [152]. Further, IL-20 expression was correlated with osteoclast activity in alveolar bone remodeling associated with OTM. Furthermore, local administration of IL-20 increased osteoclast activity and the OTM distance in rats, with these phenomena being reversed by anti-IL-20 antibody [153].

### 2.16. IL-23

IL-23 promotes the proliferation of Th17 cells, which express IL-17A, IL-17F, IL-22, IL-26, IFNγ, and TNF-α. Further, IL-23 signaling is significantly involved in the progression of chronic human diseases [154]. Specifically, IL-23 is involved in the pathogenesis of inflammatory arthritis, such as psoriatic arthritis, ankylosing spondylitis, and rheumatoid arthritis, with IL-23 levels in synovial fluid and serum being positively correlated with the severity and disease activity of rheumatoid arthritis. Furthermore, single-nucleotide polymorphisms in genes encoding the IL-23 and IL-23 receptor have been implicated in spondyloarthritis, such as psoriatic arthritis and ankylosing spondylitis [155,156]. IL-23 increases osteoclast formation in mice with collagen-induced arthritis [157].

Patients subjected to orthodontic force showed significantly higher IL-23 levels in the GCF at both the tension and pressure areas than controls. IL-23 levels are positively correlated with RANKL expression [139].

### 2.17. IL-27

IL-27 is a heterodimer that belongs to the IL-12 cytokine family and is composed of two subunits, IL-27p28 and Epstein–Barr virus-induced gene 3. IL-27 and its receptor are expressed on immune cells, including B cells, T cells, monocytes, macrophages, dendritic cells, and neutrophils, as well as non-immune cells, such as cardiac Sca-1 positive cells and renal tubular epithelial cells. The IL-27 receptor complex comprises glycoprotein 130 and IL-27ra. IL-27 exhibits both anti-inflammatory and pro-inflammatory effects. Regarding anti-inflammatory effects, IL-27 suppresses the proliferation and apoptosis of B cells, macrophages, monocytes, and dendritic cells. Contrastingly, IL-27 has been shown to exert pro-inflammatory effects through dendritic cells and various effector Th cells. Furthermore, given the relationship of IL-27 with inflammatory autoimmune diseases, IL-27 may not only inhibit autoimmunity development but also promote the pathogenesis of autoimmune diseases [158,159]. IL-27 has been shown to inhibit lipopolysaccharide-induced osteolysis in vivo. Notably, IL-27 inhibited RANKL-induced osteoclast formation by inhibiting NF-κB p65 and IκB phosphorylation [160]. Administration of exogenous IL-27 reduced soft tissue abscesses and peri-implant bone resorption during infection; however, this effect was observed in wild-type mice but not in IL-27Rα-/- mice. These results suggest that IL-27 is not essential for immunity but rather mediates redundant immune and bone cell functions during infection [161]. IL-27 significantly inhibits cell surface expression of RANKL and secretion of soluble RANKL by naive CD4+ T cells stimulated by T cell receptor engagement [162].

Compared with controls, orthodontic patients showed significantly higher IL-27 levels in GCF at both tension and compression sides in the early treatment stages; however, there was a negative correlation between IL-27 and RANKL expression during the later stages [139].

### 2.18. IL-33

There has been increasing evidence regarding the function of the IL-1 superfamily cytokine and damage-associated molecular patterns of IL-33. ST2, which is the receptor for IL-33, is expressed on the surface of various cells. IL-33 is a powerful driver of the type 2 immune response that promotes parasite clearance; moreover, it is involved in inflammatory diseases such as allergies and asthma. At steady state, full-length IL-33 is constitutively expressed in various cell types in human and mouse tissues and is localized to the nucleus. IL-33 has been shown to be produced by endothelial cells of the human vasculature; adventitial stromal cells of mice; epithelial cells of barrier tissues; fibroblastic reticular cells of lymphoid organs; and neurons, glial cells, and astrocytes of the nervous system [163]. IL-33 activation promotes the production of pro-inflammatory cytokines and induces osteoclast formation in periodontitis [164]. IL-33 inhibits RANKL-induced osteoclast formation by regulating Blimp-1 and IRF-8 expression in vitro [165]. In addition, IL-33 inhibited IκB phosphorylation and NF-κB nuclear translocation during TNF-α-induced osteoclast formation and bone resorption [166].

In a mouse model of OTM, IL-33 injection inhibited osteoclast differentiation induced by RANKL from bone marrow stromal cells in vitro. Further, it reduced the number of TRAP-positive cells in the PDL during OTM [167]. High orthodontic stress induces IL-33 expression in periodontal tissues, with IL-33 exerting a negative effect on cementum formation in mice by suppressing cementoblast differentiation, mineralization, and the expression of cementogenesis-related proteins [168].

### 2.19. IL-34

IL-34 is a pro-inflammatory cytokine that binds to four receptors (CSF-1R, syndecan-1, PTP-ζ, and TREM2). It was first identified as a ligand for the M-CSF receptor. Although there is no sequence homology between IL-34 and CSF-1, they have similar active regions. IL-34 is crucially involved in the onset and progression of various inflammatory diseases by binding to the CSF-1 receptor at the gap between the immunoglobulin-like structural domains D2 and D3 [169,170]. IL-34 inhibited RANKL-induced osteoclast formation in vitro. Moreover, local injections of mouse recombinant IL-34 significantly increased the number of osteoclasts, enhanced alveolar bone loss, and elevated cathepsin K activity in a mouse model of ligature-induced periodontitis. Notably, anti-IL-34 neutralizing monoclonal antibodies reduced the number of osteoclasts and attenuated alveolar bone loss in periodontitis lesions [171]. Serum IL-34 levels are associated with increased disease severity in rheumatoid arthritis. IL-34 is an endogenous factor that repopulates hypermetabolic M34 macrophages and promotes cross-regulation with effector T cells to promote inflammatory bone resorption in rheumatoid arthritis [172]. Wistar rats subjected to orthodontic force showed a significant increase in IL-34-positive odontoclasts during OTM [135].

### 2.20. IFN-γ

IFN-γ inhibits viral replication in response to stimulation with phytohemagglutinin. It was first identified as a soluble factor secreted by human leukocytes. IFN-γ has antiviral and immunomodulatory properties. IFN-γ directly inhibits several steps of the viral life cycle, including host cell entry, viral gene replication, and viral gene transcription, across several cell types. Regarding immune regulation, IFN-γ promotes Th1 responses in viral infections. Additionally, IFN-γ activates macrophages by stimulating nitric oxide release, enhancing production of reactive oxygen species, and increasing phagocytic activity. Furthermore, IFN-γ promotes antigen presentation and increases the expression of MHC I and II, which are antigen-presenting molecules, on antigen-presenting cells. IFN-γ, which is mainly produced by NK cells and T cells, induces immune cell proliferation and activation, cytokine expression, maturation, and effector function; additionally, it induces tumor growth arrest and apoptosis [173,174,175,176]. IFN-γ is crucially involved in bone metabolism. Specifically, IFN-γ promotes osteoblast differentiation and inhibits bone marrow adipocyte formation. The role of IFN-γ is dependent on the stage of osteoclast formation. In addition, IFN-γ is crucially involved in the pathogenesis of bone diseases, including postmenopausal osteoporosis, rheumatoid arthritis, and acquired immune deficiency syndrome. IFN-γ can enhance local inflammation in gingival tissues. IFN-γ promotes alveolar bone loss and osteoclast differentiation. The truncated tryptophanyl-tRNA synthetase-dependent action of IFN-γ promotes multinucleation of myeloid lineage, which is a crucial step in osteoclastogenesis. IFN-γ directly inhibits TNF-α-induced osteoclast formation in vitro and in vivo and induces apoptosis via the interaction of TNF-α-induced Fas in adherent cells with IFN-γ-induced Fas ligand in non-adherent cells in bone marrow cell culture [177,178,179,180].

We previously showed increased local IFN-γ mRNA expression in PDL during OTM in mice. IFN-γ-administrated mice showed lower tooth movement and osteoclast numbers on the pressure side than phosphate-buffered saline (PBS)-treated control mice. IFN-γ, which is increased in PDL during OTM, may inhibit osteoclast formation and tooth movement induced by mechanical loading [181]. Patients with grade C periodontitis showed decreased IFN-γ levels in GCF and plasma during orthodontic treatment, which may reduce the proportion of IFN-γ-positive Th1 cells [182]. IFN-γ-treated rats whose mandibular first molars were proximally moved using Ni-Ti sealed coil springs showed an increase in Tr.N and BV/TV, as well as a decrease in Tr.Sep, indicating an anti-bone resorption effect of IFN-γ [183]. Tooth movement was significantly reduced in T cell-deficient immunodeficient mice. Wild-type (WT) mice showed an increased number of TRAP-positive cells around the alveolar bone after OTM; however, this was not observed in immunodeficient mice. Taken together, T cells are required for OTM in a manner dependent on Th1-related cytokines such as IFN-γ [184]. Rats administered substance P showed a significant increase in tooth movement, as well as activated osteoclast and osteoblast activity. Finally, IFN-γ levels in peripheral blood showed an increase and decrease in periodontal tissues at the early and later stages, respectively. This demonstrates that systemic administration of substance P accelerates OTM [185].

### 2.21. TNF-α

TNF-α is an important regulator of inflammatory responses. This cytokine exerts pleiotropic effects on various cell types; further, it has been implicated in the pathogenesis of several inflammatory and autoimmune diseases [186,187]. Structurally, TNF-α is a homotrimeric protein and is primarily produced by activated NK cells, T cells, and macrophages [188,189]. Inflammation through TNF-α-mediated cell death may be beneficial in infection by providing cell-extrinsic signals that help elicit an appropriate immune response [187,190]. Functionally, it triggers the induction of various inflammatory molecules, including other cytokines and chemokines [191]. TNF-α exists in two forms, soluble and transmembrane. Transmembrane TNF-α is initially synthesized, followed by processing by TNF-α-converting enzyme, which is a membrane-bound disintegrin metalloprotease, to release soluble TNF-α [192]. TNF-α mediates osteoclast formation in vitro and in vivo [193]. TNF-α-induced osteoclast recruitment is crucially involved in the pathogenic process of inflammatory diseases [191,194]. TNF-α has been implicated in postmenopausal osteoporosis [195], periodontal diseases [196], and rheumatoid arthritis [197]. There are two cell surface receptors for TNF-α, including TNF receptor type 1 (TNFR1) and TNF receptor type 2 (TNFR2). Binding to these receptors induces biological responses, with each receptor mediating distinct intracellular signals [198]. In vitro analysis of osteoclast formation from bone marrow cells obtained from TNFR1- or TNFR2-deficient mice revealed that TNFR1 and TNFR2 induced and inhibited osteoclast differentiation, respectively [198,199].

OTM increases TNF-α levels in the human gingival sulcus [102,200,201,202,203,204,205,206]. TNF-α is expressed in the periodontal tissues of rats under pathological conditions caused by excessive orthodontic forces [207]. Upon application of tooth movement appliances, TNFR2-deficient mice showed less amount of tooth movement than wild-type mice [208]. These results suggest that TNFR2 is crucially involved in OTM. TNFR1-deficient mice had fewer osteoclasts than wild-type mice [209]. Mice without both TNFR1 and TNFR2 (TNFRsKO) showed a significant reduction in tooth movement, which further confirms the role of TNFRs [210]. A previous study performed OTM with or without injection of docosahexaenoic acid (DHA) in the local site of the gingiva around the mouse tooth; subsequently, the effects of DHA on osteoclast formation at the pressure side of the alveolar bones were examined. GPR120 is a DHA receptor and a lipid sensor. In this study, DHA suppressed OTM in WT mice but not in GPR120-deficient mice. This indicates that DHA attenuates TNF-α-induced osteoclast formation and bone resorption via GPR120. This suggests that TNF-α is crucially involved in OTM. Taken together, DHA may inhibit TNF-α-induced osteoclast recruitment and bone resorption during OTM [211]. After mesial movement of first molars, WT mice showed a significantly higher number of RANK-positive cells on the compression side than TNFRsKO mice. TNF-α directly induces RANKL expression in osteocytes and promotes osteoclast formation [212,213]. The results suggested that TNF-α induces RANK expression in vitro and at baseline in vivo, as well as RANK expression on the compression side during OTM [214]. WT mice showed a significantly higher number of RANKL-positive osteocytes in the alveolar bone after OTM than TNFRsKO mice, which suggests that TNF-α induces RANKL expression in osteocytes during OTM [215]. Compared with 8-week-old mice, 78-week-old mice showed reduced TNF-α-induced bone resorption, osteoclast formation, and calvarial expression of TRAP and cathepsin K. Furthermore, 78-week-old mice showed reduced osteoclast formation and reduced OTM, with the latter being attributed to reduced TNF-α-induced osteoclast formation [216]. Compared with WT mice, TNFRsKO mice showed fewer VEGF-positive cells in the mesial periodontal membrane of the distal buccal root during OTM, as well as reduced VEGF mRNA expression. Taken together, these results suggest that TNF-α plays a crucial role in VEGF expression during OTM [217]. Osteocyte death in TNFRsKO mice was reduced during OTM. Necroptosis markers in osteocytes on the compression side in WT mice were detected, whereas such osteocytes were almost undetectable in TNFRsKO mice. Moreover, the conditioned medium from osteocytes undergoing necroptosis significantly enhanced osteoclast formation. Taken together, TNF-α-induced osteocyte necroptosis promotes osteoclast formation and alveolar bone resorption on the compression side during OTM, and is accompanied by the release of inflammatory factors such as damage-associated molecular patterns (DAMPs) [218].

### 2.22. M-CSF

M-CSF is a protein constitutively produced by various cells, including macrophages, osteoblasts, lymphocytes, monocytes, endothelial cells, fibroblasts, and osteocytes. It was first identified as a growth factor of hematopoietic cells that induces the differentiation of bone marrow progenitor cells into macrophages [219]. M-CSF is essential for the proliferation and differentiation of osteoclast precursor cells; moreover, it induces the proliferation, differentiation, and survival of macrophages, monocytes, and bone marrow precursor cells [220]. In osteoporotic (op/op) mice, M-CSF deficiency caused by thymidine insertion into the Csf-1 gene reduces osteoclast and macrophage function [221]. Therefore, M-CSF is considered an essential factor for osteoclast formation and activation [222]. Furthermore, administration of the M-CSF receptor c-fms antibody completely inhibited osteoclast formation as well as TNF-α-induced bone destruction and inflammatory arthritis [223]. Furthermore, neutralizing anti-c-fms antibody suppressed osteoclast formation and bone loss compared with that in PBS-treated ovariectomized mice [224].

Local injection of anti-c-fms antibody during mechanical loading significantly inhibited OTM and significantly reduced the number of osteoclasts [210]. Furthermore, M-CSF injection promoted OTM and osteoclast formation [225]. Notably, administration of anti-c-fms antibodies inhibited osteoclast formation in mice as well as reduced orthodontic relapse after OTM [226].

### 2.23. TGFs

There are two main types of TGFs: TGF-α and TGF-β. The TGF-β superfamily is a large and highly diverse group of structurally related proteins that act as cytokines and growth factors. The BMP subfamily is within the broader TGF-β superfamily. TGF-β regulates both innate and adaptive immune responses. Further, it has several biological functions in cell biology, cancer biology, and the immune response [227,228]. TGF-β is recognized as a key cytokine with a dual role in both immunity and tolerance. TGF-β was originally considered an immunomodulatory cytokine since it inhibits inflammatory cytotoxic cells and Th cells, as well as promotes immunosuppressive Treg cells. However, it was subsequently found to induce Th17 cells, which have both pathogenic and immunoregulatory functions [229]. Another study showed that low TGF-β levels promote osteoclast differentiation, with both M-CSF levels and the RANKL/OPG ratio being higher. Contrastingly, high TGF-β levels suppress the RANKL/OPG ratio since TGF-β inhibits RANKL expression and enhances OPG expression in osteoblasts [230]. Combined with dose-dependent inhibition of M-CSF expression by TGF-β, osteoclast formation is inhibited [231]. B cells have been shown to secrete TGF-β, which inhibits osteoclast formation and shortens the lifespan of mature osteoclasts [232]. Osteoclastogenesis inhibition by TGF-β is primarily achieved by reducing RANKL secretion by osteoblasts; contrastingly, TGF-β significantly increased osteoclastogenesis in hematopoietic cell cultures stimulated with recombinant RANKL or TNF-α [233]. It has been reported that TGF-α increased osteoclast formation by stimulating osteoclast precursors in human marrow culture [234]. The dual effect of BMPs on bone resorption and mineralization highlights the essential role of BMP signaling in bone homeostasis. BMPs are well known to induce osteoblast formation and bone formation. Furthermore, BMPs also induced osteoclast formation and bone resorption [235].

A human study on tooth extraction following OTM showed increased TGF-β expression in both the tension and compression areas [13]. Further, increased TGF-β levels in GCF have been reported during OTM in humans [236]. Notably, low-level laser therapy in OTM did not increase TGF-β1 levels [36]. Bioinformatics analysis of gingival crevicular fluid during OTM demonstrated increased TGF-β levels [237]. In a rat model of OTM, orthodontic force and assisted vibration were applied to the maxillary right first molar, followed by injection of the TGF-β receptor inhibitor SB431542 into the palatal and buccal submucosa. SB431542 inhibited the vibration-induced promotion of tooth movement and an increase in the number of osteoclasts. Further, assisted vibration increased the number of TGF-β-positive bone cells in the compressed alveolar bone during OTM. Taken together, assisted vibration promoted OTM by promoting the production of TGF-β in bone cells and subsequent osteoclast formation [238]. Rats injected with platelet-rich plasma showed less tooth movement compared to the control group on day 3. However, all groups showed maximum tooth movement on day 14, with no significant between-group differences. Further, there were no significant between-group differences in the number of osteoclasts and osteoblasts, as well as in TGF-β, ALP, and TRAP expression [239]. It has been reported that the expression of BMPs increased on the tension side during OTM, stimulating the differentiation of mesenchymal stem cells to osteoblasts [240]. BMP-3 expression is gradually increased on the tension side until day 14 in rodent models of OTM, the mid-stage in OTM [241]. One study examined the effect of BMP2 injection on the pressure and tension side of orthodontic tooth in rats and found that local injection of BMP-2 inhibited OTM. BMP-2 enhanced osteoclast formation, although bone resorption was not dominant during OTM [242].

### 2.24. RANKL and OPG

RANKL is also termed the osteoclast differentiation factor, TNF-related activation-inducing cytokine, TNF ligand superfamily member 11, and OPG ligand [243]. RANKL is highly produced in activated T lymphocytes, osteoblasts, and osteocytes [244]. RANK has extracellular and intracellular domains, with each containing four cysteine-rich pseudo-repeats at the amino terminus and three TRAF-binding domains at the carboxy terminus [245]. RANK is primarily expressed by osteoclast precursors, mature osteoclasts, and immune cells such as dendritic cells, microglia, and macrophages [246]. Signaling by RANKL-RANK binding activates osteoclast differentiation and function, inhibits osteoclast apoptosis, and induces bone resorption, while the RANKL decoy receptor OPG inhibits RANKL-RANK binding [222]. Osteocytes, which act as mechanosensors, are an important source of RANKL [247]. Notably, OPG expression by osteocytes is significantly reduced in osteocyte-specific B-catenin knockout mice. Additionally, osteocyte OPG-knockout mice exhibit an osteopenic phenotype, indicating that OPG expression in osteocytes is crucially involved in regulating bone mass [248].

RANKL-injected rats showed enhanced OTM, which was consistent with a significant increase in TRAP activity [249]. Similarly, local injection of RANKL in mice increases osteoclast activity and promotes tooth movement [250]. In humans, RANKL and OPG levels in GCF were higher and lower, respectively, in OTM teeth than in control teeth [251]. Similar findings were reported in a rat OTM model, in which two different orthodontic forces were applied to the bilateral buccal expansions around the maxillary second and third molars. The low-force group showed increased sRANKL expression and sRANKL/OPG ratio compared with the normal-force and control groups [53]. Compared with WT mice, OPG-deficient mice exhibited greater tooth movement distances, enhanced alveolar bone resorption, greater osteoclast numbers, and higher serum TRAP levels [252]. Osteocyte-derived RANKL promotes osteoclast formation during OTM [215]. Osteocyte-specific RANKL-deficient mice showed reduced OTM due to inhibition of osteoclast formation in periodontal tissue [253].

## 3. Summary and Limitations

During OTM, various cytokines play regulatory roles—some promote the process, while others inhibit it. Cytokines such as IL-1, IL-2, IL-6, IL-8, IL-10, IL-11, IL-13, IL-17, IL-18, IL-20, IL-23, IL-27, IL-33, IL-34, IFN-γ, TNF-α, M-CSF, TGF-β, RANKL, and OPG have been identified during OTM through analyses of GCF, immunostaining, and RNA expression. Among these, IL-1, IL-6, IL-20, TNF-α, M-CSF, TGF-β, and RANKL enhance OTM, while IL-4, IL-12, IL-33, IFN-γ, and OPG suppress it. These findings are based on studies involving cytokine administration, cytokine-neutralizing antibody administration, and cytokine-deficient mice. Some of the cytokines discussed in this review promote osteoclastogenesis, including IL-1, IL-6, IL-7, IL-8, IL-11, IL-16, IL-17, IL-20, IL-23, TNF-α, M-CSF, TGF-β, and RANKL. Others inhibit osteoclastogenesis, such as IL-4, IL-5, IL-10, IL-12, IL-13, IL-18, IL-27, IL-33, IL-34, IFN-γ, TGF-β, and OPG. These observations suggest that cytokines that enhance osteoclast formation may promote OTM, while those that inhibit osteoclastogenesis may suppress it (Figure 2).

First, there is a limitation in research on the expression of cytokines depending on the strength of the orthodontic force. There are differences in the strength, application method, and duration of the orthodontic force in experimental animals such as mice, rats, rabbits, and dogs. These variables have not been standardized across studies. Therefore, it is unclear how these cytokines interact with each other during different stages of OTM. It is necessary to standardize the animal species, the method of applying the orthodontic force, and the duration, and to examine the expression and effects of cytokines by varying the strength of the orthodontic force. Furthermore, understanding how these cytokines interact with each other during different stages of OTM may provide insight into identifying which of the cytokines to serve as biomarkers in GCF to monitor the progress of orthodontic treatment.

The main limitation of this review is the lack of studies directly investigating the effects of individual cytokines on OTM in humans. Most of the studies that investigated how cytokines affect OTM in this review were animal studies, which allow for experimental manipulations such as administering cytokines, administering antibodies that suppress cytokines, or the use of cytokine-deficient mice. Therefore, studies that directly investigate the effects of cytokines on OTM in humans are also needed. In addition, even in human studies of cytokine expression during orthodontic treatment, there is still variability due to differences in methods used, such as the method of orthodontic force application, location, duration of force application, and patient age. To address this issue, human studies with more standardized and controlled methodologies are needed.

Recent research indicates that epigenetic mechanisms significantly regulate cytokine expression and cellular responses within the PDL during OTM. The differentiation and function of periodontal tissue cells, such as periodontal ligament cells, osteoblasts, osteoclasts, as well as cytokine expression, may be regulated by epigenetic mechanisms, which include histone modification and non-coding RNA [105,254,255]. In particular, the importance of epigenetic regulation in stem cell differentiation and bone homeostasis has been recognized, and these mechanisms are thought to be involved in cytokine expression during OTM [256]. Research into epigenetic regulation of cytokines during OTM will be of great significance in promoting a deeper understanding of the mechanism of tooth movement and improving the efficiency, predictability, and safety of orthodontic treatment. Therefore, epigenetic regulation represents an important direction for future research (Table 1).

## 4. Conclusions

This review discusses the various cytokines involved in OTM, either by suppressing or promoting tooth movement. The influence of cytokines on OTM involves the formation of osteoclasts; i.e., cytokines that inhibit and promote osteoclast formation probably inhibit and promote OTM, respectively. These cytokines are expressed by various cells during OTM; moreover, they act on bone-related cells such as osteoblasts, osteoclasts, and osteocytes. Further studies, including single-cell sequencing analysis, are warranted to further elucidate the role of cytokines in OTM.

## Figures and Tables

**Figure 1 ijms-26-06688-f001:**
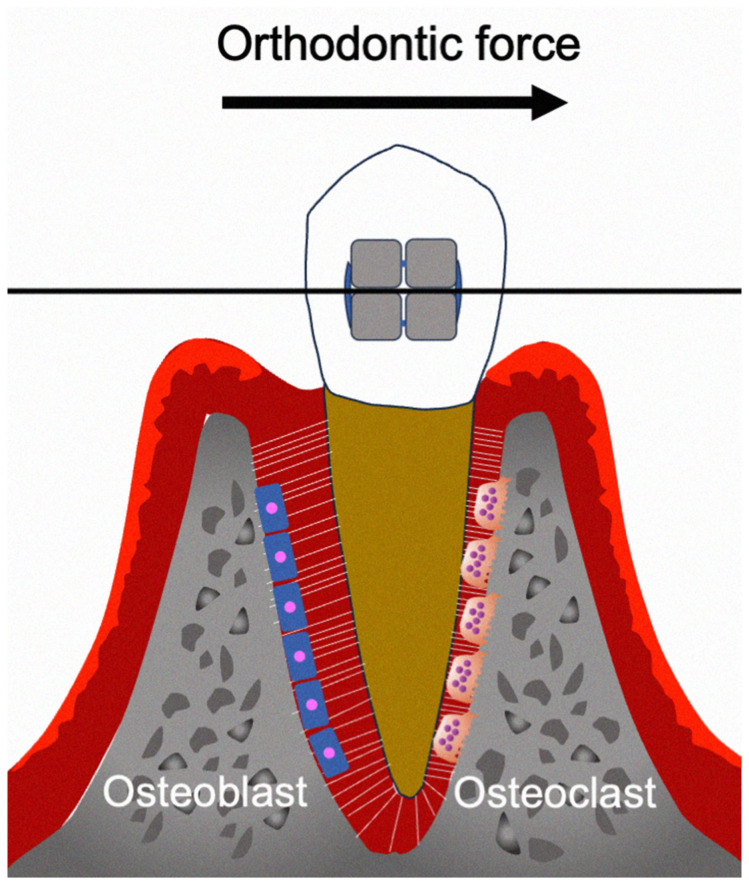
Schema of the biological effects during OTM.

**Figure 2 ijms-26-06688-f002:**
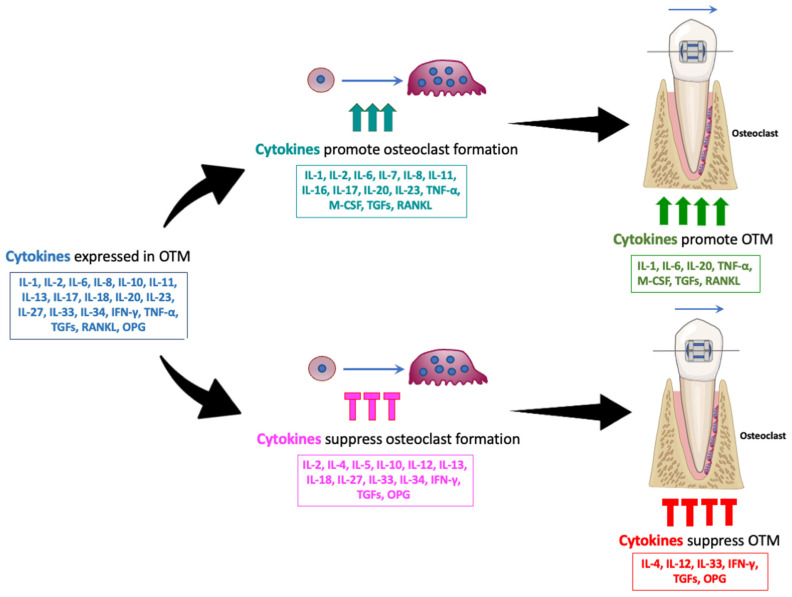
Various cytokines influence OTM, with some promoting and others suppressing the process. Multiple cytokines, including IL-1, IL-2, IL-6, IL-8, IL-10, IL-11, IL-13, IL-17, IL-18, IL-20, IL-23, IL-27, IL-33, IL-34, IFN-γ, TNF-α, M-CSF, TGFs, RANKL, and OPG, have been identified during OTM through analyses of GCF, immunostaining, and RNA expression. Cytokines such as IL-1, IL-6, IL-7, IL-8, IL-11, IL-16, IL-17, IL-20, IL-23, TNF-α, M-CSF, TGFs, and RANKL promote osteoclastogenesis. Others, including IL-4, IL-5, IL-10, IL-12, IL-13, IL-18, IL-27, IL-33, IL-34, IFN-γ, TGFs, and OPG, inhibit osteoclastogenesis. Among these, IL-1, IL-6, IL-20, TNF-α, M-CSF, TGF-β, and RANKL were reported to promote OTM, while IL-4, IL-12, IL-33, IFN-γ, and OPG were reported to suppress it. These findings suggest that cytokines that enhance osteoclast formation may promote OTM, while those that inhibit osteoclastogenesis may suppress it.

**Table 1 ijms-26-06688-t001:** Effects of cytokines on osteoclast formation and orthodontic tooth movement.

Cytokines	Effect on Osteoclast Formation	Source (Osteoclast)	Detection of Cytokines During OTM	Effect on OTM	Source (OTM)
**IL-1β**	+	[6,19,20,21,22,23]	detect	+	[25,26,28,29,30,31,32,33,34,35,36,37,38]
**IL-2**	+ or −	[39] (+), [40] (−)	detect		[42]
**IL-4**	−	[46,47,48,49]		−	[50,51,53]
**IL-5**	−	[55]	undetectable		[56]
**IL-6**	+	[21,64,65,66,67]	detect	+	[35,68,69,70,71,72,73,74,75]
**IL-7**	+	[77,78,79,80]	no different		[81]
**IL-8**	+	[87]	detect		[36,71,88,89,90,91,92]
**IL-10**	−	[96,97,98,99,100,101]	detect		[102,103,104]
**IL-11**	+	[108,109,110,111]	detect		[112,113]
**IL-12**	−	[118,119,120]		−	[121]
**IL-13**	−	[125,126]	detect		[102]
**IL-16**	+	[128]	no different		[129]
**IL-17**	+	[133,134]	detect		[135,136,137,138,139]
**IL-18**	−	[120,145,146,147]	detect		[34,113,148]
**IL-20**	+	[150,151]	detect	+	[152,153]
**IL-23**	+	[157]	detect		[139]
**IL-27**	−	[160]	detect		[139]
**IL-33**	−	[165,166]	detect	−	[167,168]
**IL-34**	−	[171,172]	detect		[135]
**IFN-γ**	−	[177,178,179,180]	detect	−	[181,182,183,184,185]
**TNF-α**	+	[191,193,194,198,199]	detect	+	[102,200,201,202,203,204,205,206,207,208,209,210,212,213,214,215,216,217,218]
**M-CSF**	+	[219,220,221,222,223]		+	[210,224,225]
**TGFs**	+ or −	[223] (+), [230,231,232,234,235] (+)	detect	+ or −	[13,236,237,238,239,240,241,242]
**RANKL**	+	[221,242,243,244,245,246]	detect	+	[53,215,248,249,250,252]
**OPG**	−	[221,247]	detect	−	[53,250,251]

+: stimulation, −: inhibition, no different: No difference from the control.

## Data Availability

Data are available from the corresponding authors upon reasonable request.

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
