# Peer review of "The Role of Cytokines in Orthodontic Tooth Movement"

_ijms, 2025, doi:10.3390/ijms26146688_

Round 1
Reviewer 1 Report
Comments and Suggestions for Authors
This manuscript provides a detailed and thorough review of cytokines related to orthodontic tooth movement (OTM). It offers a clear classification of both pro- and anti-inflammatory cytokines and explains their molecular pathways. The paper is a valuable contribution to the understanding of immune mechanisms in OTM and will be useful to both clinicians and researchers in orthodontics and molecular biology. I recommend acceptance after minor revisions.
Comments:
- While the paper explains individual cytokines and their roles in osteoclast activity, it does not clearly describe how these cytokines interact with each other during different stages of OTM (such as initiation, peak, and resolution). A figure or summary table may help organize and present this information more clearly.
- Among the cytokines discussed, which ones show potential as biomarkers in gingival crevicular fluid (GCF) to monitor the progress of orthodontic treatment?
- Some sections, especially those covering IL-4 to IL-23, tend to be repetitive in format and do not provide much analysis. The authors might consider summarizing these in a table and focusing the main text on unique or conflicting findings.
- There are minor grammar and phrasing issues throughout the manuscript. For example, the sentence "It important to understand..." should be corrected to "It is important to understand...". A careful proofreading is advised.
- The manuscript does not discuss epigenetic control of cytokine expression during OTM. Including a short section on this topic would strengthen the paper.
Author Response
To Reviewer 1:
Thank you very much for your constructive comments and suggestions. I sincerely appreciate them and will answer them in order.
While the paper explains individual cytokines and their roles in osteoclast activity, it does not clearly describe how these cytokines interact with each other during different stages of OTM (such as initiation, peak, and resolution). A figure or summary table may help organize and present this information more clearly.
There are differences in the strength, method, and duration of the orthodontic force in experimental animals such as mice, rats, rabbits, and dogs, and the researches have not been normalized. Therefore, it is also unclear how these cytokines interact with each other during different stages of OTM. As a limitation, we added the following: "First, there is a limitation in research on the expression of cytokines depending on the strength of the orthodontic force. There are differences in the strength, application method, and duration of the orthodontic force in experimental animals such as mice, rats, rabbits, and dogs. These variables have not been standardized across studies. Therefore, it is unclear how these cytokines interact with each other during different stages of OTM. It is necessary to standardize the animal species, the method of applying the orthodontic force, and the duration, and to examine the expression and effects of cytokines by varying the strength of the orthodontic force." in Summary and limitation section.
Among the cytokines discussed, which ones show potential as biomarkers in gingival crevicular fluid (GCF) to monitor the progress of orthodontic treatment?
From previous reports, it is unclear which of the cytokines to serve as biomarkers in gingival crevicular fluid (GCF) to monitor the progress of orthodontic treatment. Therefore, we added the following: “Furthermore, understanding how these cytokines interact with each other during different stages of OTM may provide insight into identifying which of the cytokines to serve as biomarkers in GCF to monitor the progress of orthodontic treatment.” in Summary and limitation section.
Some sections, especially those covering IL-4 to IL-23, tend to be repetitive in format and do not provide much analysis. The authors might consider summarizing these in a table and focusing the main text on unique or conflicting findings.
I made a table to make it easier to understand.
There are minor grammar and phrasing issues throughout the manuscript. For example, the sentence "It important to understand..." should be corrected to "It is important to understand...". A careful proofreading is advised.
We checked grammar and phrasing issues, and corrected them.
The manuscript does not discuss epigenetic control of cytokine expression during OTM. Including a short section on this topic would strengthen the paper.
Regarding epigenetic control of cytokines in OTM, this is a large field in itself, and we felt that including it in this paper would be loss of the focus, so we decided to state the following: “Recent research indicates that epigenetic mechanisms significantly regulate cytokine expression and cellular responses within the PDL during OTM. The differentiation and function of periodontal tissue cells such as periodical ligament cells, osteoblasts, osteoclasts, as well as cytokine expression, may be regulated by epigenetic mechanisms which are histone modification and non-coding RNA [105,254,255]. In particular, the importance of epigenetic regulation in stem cell differentiation and bone homeostasis has been recognized, and these mechanisms are thought to be involved in cytokine expression during OTM [256]. Research into epigenetic regulation of cytokines during OTM will be of great significance in promoting a deeper understanding of the mechanism of tooth movement and improving the efficiency, predictability, and safety of orthodontic treatment. Therefore, epigenetic regulation represents an important direction for future research.” in Summary and limitation section.

Reviewer 2 Report
Comments and Suggestions for Authors
Dear authors, the chosen topic is topical and of great interest.
Please, please:
-what were the criteria followed for the inclusion of studies?
-which of the authors analysed the articles?
- what was the intensity of orthodontic forces?
Thank you.
Author Response
To Reviewer 2:
Thank you very much for your constructive comments and suggestions. I sincerely appreciate them and will answer them in order.
-what were the criteria followed for the inclusion of studies?
First, we searched for tooth movement and each cytokine in PubMed, and narrowed down the search to cytokines that had been published in papers. Next, we searched for those cytokines and osteoclasts in PubMed. When citing papers, we tried to avoid papers older than 5 years as much as possible. As a result, papers published within the past 5 years accounted for more than 50% of the total.
-which of the authors analysed the articles?
First author Kitaura analyzed the articles.
- what was the intensity of orthodontic forces?
The intensity of orthodontic force varies, and it also differs depending on the animal, so the strength and expression of cytokines are not clear. As a limitation, we added the following: "First, there is a limitation in research on the expression of cytokines depending on the strength of the orthodontic force. There are differences in the strength, application method, and duration of the orthodontic force in experimental animals such as mice, rats, rabbits, and dogs. These variables have not been standardized across studies. Therefore, it is unclear how these cytokines interact with each other during different stages of OTM. It is necessary to standardize the animal species, the method of applying the orthodontic force, and the duration, and to examine the expression and effects of cytokines by varying the strength of the orthodontic force." in Summary and limitation section.

Reviewer 3 Report
Comments and Suggestions for Authors
IJMS-3701890: Review:
The Role of Cytokines in Orthodontic Tooth Movement, by Hideki Kitaura et al.
GENERAL COMMENTS:
The authors reviewed on the roles and actions of various cytokines in the orthodontic tooth movement (OTM) and related immune and inflammatory reactions. They introduced the cytokine-induced various cellular responses, including cell proliferation, differentiation, cell death, and functional expression. This review contains some interests for the specific readers; however, the composition of the review and the discussion on the interactions between the cytokines and the relationship to the pathophysiological conditions should be improved.
SPECIFIC COMMENTS:
1) Before the explanation of each cytokine, the lineup of all the players involved in the OTM should be introduced and the hierarchy of each cytokine should also be discussed.
2) The interactions and interrelationships between the related cytokines related to the OTM process should be graphically shown as a figure.
3) The key actions and roles of each cytokine should be summarized in a Table.
4) In the section of TGFs, the activity of TGF-alpha and the subfamily of BMPs need to be introduced.
5) The authors mentioned only about the effects as cytokine ligands in each section. The explanation regarding the functional receptors, the structure for individual cytokines, and their intracellular signaling downstream of each receptor should be discussed.
6) Fig. 2 is too simple to explain the roles of each cytokine on the process of OTM. Much clearer and attractive diagram and the explanation based on the texts should be needed.
Comments on the Quality of English Language
Needs native checkup.
Author Response
To Reviewer 3:
Thank you very much for your constructive comments and suggestions. I sincerely appreciate them and will answer them in order.
1) Before the explanation of each cytokine, the lineup of all the players involved in the OTM should be introduced and the hierarchy of each cytokine should also be discussed.
I added the lineup of all the cytokines and introduced them as following “In this review, we described the characteristics of the following cytokines; L-1, IL-2, IL-4, IL-5, IL-6, IL-7, IL-8, IL-10, IL-11, IL-12, IL-13, IL-16, IL-17, IL-18, IL-20, IL-23, IL-27, IL-33, IL-34, IFN-γ, TNF-α, M-CSF, TGF-β, RANKL, and OPG, their relationship to immune system, bone metabolic diseases, their effects on osteoclast formation, and their effects on OTM.”.
We add table of the key actions and roles of each cytokine as hierarchy of each cytokine.
2) The interactions and interrelationships between the related cytokines related to the OTM process should be graphically shown as a figure.
We have updated Fig. 2 to include more details. And we added explanation based on the text in Figure 2 legend. I think if we add figure which is all the interactions and interrelationships between cytokines related to the OTM, it would be very complicated.
3) The key actions and roles of each cytokine should be summarized in a Table.
I made a table to make it easier to understand.
4) In the section of TGFs, the activity of TGF-alpha and the subfamily of BMPs need to be introduced.
We added about TGF-α and about the BMP subfamily as following “There are two main types of TGFs: TGF-α and TGF-β. TGF-β superfamily is a large and highly diverse group of structurally related proteins that act as cytokines and growth factors. The BMP subfamily is within the broader TGF-β superfamily.”. We also added about effect of TGF-α and BMPs to bone metabolism as following “It has been reported that TGF-α increased osteoclast formation by stimulating osteoclast precursors in human marrow culture [234]. The dual effect of BMPs on bone resorption and mineralization highlights the essential role of BMP-signaling in bone homeostasis. BMPs are well known to induce osteoblast formation and bone formation. Furthermore, BMPs also induced osteoclast formation and bone resorption [235].”. Finally, we added the effect of BMPs during OTM as following “It has been reported that the expression of BMPs increased on the tension side during OTM, stimulating the differentiation of mesenchymal stem cells to osteoblasts [240]. BMP-3 expression is gradually increased on the tension side until day 14 in rodent models of OTM, the mid-stage in OTM [241]. One study examined the effect of BMP2 injection on pressure and tension side of orthodontic tooth in rats and found that local injection of BMP-2 inhibited OTM. BMP-2 enhanced osteoclast formation although bone resorption was not dominant during OTM [242].”.
5) The authors mentioned only about the effects as cytokine ligands in each section. The explanation regarding the functional receptors, the structure for individual cytokines, and their intracellular signaling downstream of each receptor should be discussed.
Regarding the functional receptors, the structure for individual cytokines, and their intracellular signaling downstream of each receptor, these are large fields in themselves, and we felt that including them in this paper would be loss of the focus, so we did not include them.
6) Fig. 2 is too simple to explain the roles of each cytokine on the process of OTM. Much clearer and attractive diagram and the explanation based on the texts should be needed.
We have updated Fig. 2 to include more detail. And we added explanation based on the text in Figure 2 legend.

Reviewer 4 Report
Comments and Suggestions for Authors
Cytokines play a crucial role in the biological processes involved in orthodontic tooth movement. When mechanical forces are applied to teeth through orthodontic appliances, a series of biological responses are triggered, leading to the remodeling of the periodontal ligament and alveolar bone. Cytokines serve as key regulators in the complex biological processes that facilitate orthodontic tooth movement by mediating inflammation, promoting bone remodeling, and coordinating tissue responses to mechanical stimuli. Understanding their roles can help improve orthodontic treatment outcomes and minimize potential side effects. Therefore, the study of Dr. Kitaura et al. is very important.
Comments
- Introduction: Here the authors should present a general description of the involvement of cytokines in OTM including 1) The response to mechanical forces when orthodontic forces are applied and the PDL is compressed on one side and stretched on the other. 2) Cytokines as key mediators of the inflammatory response which help to recruit immune cells to the site of tooth movement, which is essential for initiating the remodeling process. 3) Cytokine importance for bone remodeling. 4) The role of cytokines in promoting tissue repair and regeneration after tooth movement. 5) Their role in modulating the expression of other factors involved in bone metabolism, such as BMPs, IGF, etc which further influence osteogenesis during orthodontic treatment. 6) The importance of timing and balance of cytokine release which are critical for effective orthodontic tooth movement. This should be corrected.
- All the typos should be corrected.
- Lines 140-141; 598-599: “between-group differences” should be substituted for “intergroup differences”.
- Lines 146-149; 301-303: These sentences are not clear. They should be clarified.
- The authors should create a Table demonstrating the data related to association of individual cytokines with OTM including the corresponding references. This should be corrected.
Author Response
To Reviewer 4:
Thank you very much for your constructive comments and suggestions. I sincerely appreciate them and will answer them in order.
1. Introduction: Here the authors should present a general description of the involvement of cytokines in OTM including 1) The response to mechanical forces when orthodontic forces are applied and the PDL is compressed on one side and stretched on the other. 2) Cytokines as key mediators of the inflammatory response which help to recruit immune cells to the site of tooth movement, which is essential for initiating the remodeling process. 3) Cytokine importance for bone remodeling. 4) The role of cytokines in promoting tissue repair and regeneration after tooth movement. 5) Their role in modulating the expression of other factors involved in bone metabolism, such as BMPs, IGF, etc which further influence osteogenesis during orthodontic treatment. 6) The importance of timing and balance of cytokine release which are critical for effective orthodontic tooth movement. This should be corrected.
Thank you for valuable comments. We add following sentences “Cytokines are central mediators of the biological cascade during OTM. When orthodontic force compresses the PDL on one side and stretches it on the other, various cytokines are secreted in response to mechanical stress [3]. Many cytokines are involved in the recruitment and activation of immune cells at the site of force application, thereby initiating and regulating bone remodeling during OTM [13,14]. They regulate osteoclast differentiation and bone resorption, with some promoting osteoclastogenesis and others acting as inhibitors [15]. Following the active phase of bone resorption, cytokines also participate in tissue repair and regeneration during OTM. Some cytokines, such as insulin-like-growth factor (IGF) and bone morphogenetic proteins (BMPs), are involved in promoting osteoblast differentiation, angiogenesis, and bone formation, thereby facilitating the remodeling of periodontal tissues [16,17]. Importantly, the timing, balance, and local concentration of cytokine release are critical to ensure efficient and controlled OTM while avoiding adverse tissue destruction. Cytokine expression is dynamic and varies during different stages of OTM, reflecting the tightly regulated interplay between inflammatory activation and resolution phases.” in introduction section.
2. All the typos should be corrected.
I checked all the typo and corrected.
3. Lines 140-141; 598-599: “between-group differences” should be substituted for “intergroup differences”.
I changed “between-group differences” to “intergroup differences”.
4. Lines 146-149; 301-303: These sentences are not clear. They should be clarified.
I deleted “Another study showed that”, and I changed sentence to “GCF samples were taken 10 weeks after initial appliance placement at 4 hours, 7 days, and 42 days after distal force was applied to the maxillary canines. IL-5 levels were measured in GCF using multiplex assays. However, the level of IL-5 was undetectable in the samples”.
I changed sentence to “IL-16 is a proinflammatory cytokine primarily known for its chemotactic properties, meaning it attracts specific immune cells including CD4+ lymphocytes, monocytes, and eosinophils, to sites of inflammation or infection”
5. The authors should create a Table demonstrating the data related to association of individual cytokines with OTM including the corresponding references. This should be corrected.
I made a table to make it easier to understand. And we added explanation based on the text in Figure 2 legend.

Round 2
Reviewer 3 Report
Comments and Suggestions for Authors
The authors have appropriately revised their manuscript according to the referee's opinion.